# Modeling Consumer Acceptance and Usage Behaviors of m-Health: An Integrated Model of Self-Determination Theory, Task–Technology Fit, and the Technology Acceptance Model

**DOI:** 10.3390/healthcare11111550

**Published:** 2023-05-25

**Authors:** Da Tao, Zhixi Chen, Mingfu Qin, Miaoting Cheng

**Affiliations:** 1Institute of Human Factors and Ergonomics, College of Mechatronics and Control Engineering, Shenzhen University, Shenzhen 518060, China; taoda@szu.edu.cn (D.T.);; 2Academy of Music, Hong Kong Baptist University, Hong Kong, China; 3Department of Educational Technology, Faculty of Education, Shenzhen University, Shenzhen 518060, China

**Keywords:** m-health, technology acceptance, task–technology fit, self-determination theory, demographic characteristics

## Abstract

Although mobile health (m-health) has great potential to reduce the cost of medical care and improve its quality and efficiency, it is not widely accepted by consumers. In addition, there is still a lack of comprehensive insight into m-health acceptance, especially among consumers with different demographic characteristics. This study aimed to explore the factors affecting consumers’ acceptance and usage behaviors of m-health and to examine whether their roles differ by demographic characteristics. A comprehensive m-health acceptance model was proposed by integrating factors from the Self-Determination Theory, Task–Technology Fit, and Technology Acceptance Model. Survey data were collected from 623 Chinese adults with at least 6 months of m-health usage experience and analyzed using structural equation modeling techniques. Multi-group analyses were performed to assess whether the model relationships were different across gender, age, and usage experience. The results indicated that relatedness and competence were significant motivational antecedents of perceived ease of use. Task–technology fit and the perceived ease of use significantly affected the perceived usefulness. The perceived ease of use and perceived usefulness were significant determinants of consumer usage behaviors of m-health and together explained 81% of its variance. Moreover, the relationships among autonomy, perceived usefulness, and usage behaviors of m-health were moderated by gender. Consumer usage behaviors of m-health were affected by factors such as self-motivation (i.e., relatedness and competence), technology perceptions (i.e., perceived ease of use and perceived usefulness), and task–technology fit. These findings provide a theoretical underpinning for future research on m-health acceptance and provide empirical evidence for practitioners to promote the better design and use of m-health for healthcare activities.

## 1. Introduction

Statistics report that current mobile Internet users account for more than half of the global population [1]. China has the largest number of mobile Internet users, with 64% of its total population accessing the mobile Internet. The proportion of mobile Internet users is even more than 80% in some developed countries, such as the USA (84%) and Japan (92%) [1]. With the proliferation of the mobile Internet, mobile health (m-health) has also rapidly expanded [2]. Through mobile Internet and mobile devices, m-health permits access to various types of powerful mobile medical services and thus has been widely recognized to have great potential to improve the effectiveness, quality, and affordability of healthcare [3]. It can integrate multiple functions to meet diverse purposes of healthcare for consumers, such as the collection and real-time monitoring of health data, provision of healthcare information and services, consultation with health professionals, and assistance with personal medical decision-making [4,5,6]. In addition, m-health is also considered an effective tool to relieve the increasing healthcare burden from the rapidly expanding elderly population and patients with chronic conditions, as it can deliver healthcare services anytime and anywhere at low cost, overcoming geographical, temporal, and even organizational barriers [7]. Given the great benefits of m-health, many countries have advocated using m-health services as a complementary approach to delivering healthcare services. The use of m-health also accelerated during the COVID-19 outbreak, as people had a strong need to avoid face-to-face contact in daily activities to prevent infections [8].

However, despite the great potential of m-health for healthcare and the national-wide policy support, its development and application have appeared stagnant in recent years, indicating a seemingly decreased acceptance by consumers [2]. Recent literature shows that one of the main reasons why m-health is often rejected or underused could be that the key antecedents for user acceptance have largely been ignored in their design and implementation [9]. For example, an international market report noted that among the 133 million diabetics who have access to diabetes apps, only 1.2% of them have used the apps for disease management [10]. Therefore, the troubling problem of non-acceptance or under-use continues to be an essential concern in m-health practice.

Indeed, a great number of studies have made efforts to explain and promote the user acceptance of m-health within the framework of varied theories [2,3,5,6,11,12,13,14,15,16,17,18,19,20,21], including some primitive social psychological theories (i.e., the Theory of Reasoned Action (TRA) [12], Theory of Planned Behavior (TPB) [11], and the Technology Acceptance Model (TAM) [3]), The Unified Theory of Acceptance and Use of Technology (UTAUT) [22], and their derivative models [23]. Among these, TAM and its extensions [19,24,25] are the most widely used due to their parsimony and effectiveness in explaining acceptance behavior. Although existing studies have examined a set of variables to facilitate m-health acceptance, such as the perceptions of m-health (e.g., perceived ease of use, perceived usefulness, and perceived privacy risk), social factors (e.g., subjective norms and social influence), and personal factors (e.g., age, gender, and experience) (see Table 1 for a summary), there is a surprising dearth of research on the motivational and technical factors that are crucial in understanding consumers’ usage of m-health. First, m-health is mainly used by younger and middle-aged consumers who are well-educated and skillful at complicated technology operations, and thus tend to dominate the interaction processes with m-health. This means that the use of m-health is largely characterized by such features as self-domination, employment of technical skills, and frequent interaction with online peers or healthcare providers [26]. The three features are well matched with the three basic self-motivation constructs of the Self-Determination Theory (SDT) of Motivation (i.e., autonomy, competence, and relatedness) [27]. The feature of self-domination indicates implications for the autonomous use of m-health to deal with healthcare issues, the feature necessitating the employment of technical skills could be considered as the competent use of m-health, and the frequent interaction feature represents the relatedness requirement in SDT. Therefore, the three self-motivation constructs in SDT may be effective to understand consumers’ usage behaviors of m-health. 

In addition, the acceptance and use of m-health could also be related to users’ initial interaction experience with the technology, which is largely determined by the fit between the functions that m-health performs and the tasks required to be completed in healthcare activities [24]. It appears that when users believe the characteristics of m-health (e.g., its functionality) cannot match their task requirements, they tend to abandon the technology. Thus, m-health relies heavily on an appropriate task–technology fit to create a good interaction experience for users, which is the core of the Task–Technology Fit (TTF) model [32]. It has been consistently documented that a poor fit between task and technology could result in interruption during technology use and probably lead to technology abandonment [33,34]. This means that task–technology fit might affect users’ behavioral responses to m-health, particularly the decision-making on m-health acceptance [24]. Therefore, we inferred that whether users accept m-health or not may not only be determined by the abovementioned motivational factors but also relies on a good task–technology fit. Although TTF and TAM have been previously used to explain acceptance in a variety of health informatics applications, their underlying theoretical assumptions are different. While TTF highlights the fit between technology characteristics (e.g., the functionality of m-health) and task requirements [32], TAM is concerned with users’ perceptions (e.g., beliefs and attitudes) about using the technology or service [35]. The results from previous studies have suggested that the combination of TAM and TTF better explains the variances in technology acceptance than TAM and TTF alone [24]. However, few studies have examined consumers’ m-health acceptance from integrated perspectives of technology perceptions, users’ self-motivations, and task–technology fit.

Moreover, previous studies on varied technologies, including automated vehicles [36], e-learning systems [37], and health informatics applications [19], have suggested that consumers’ technology acceptance might vary across different demographic characteristics. For example, the relationships between the determinants of technology acceptance and intention to use smart healthcare services could be moderated by age, gender, and previous experience [5,38]. Nevertheless, how the relationships in the integrated framework of SDT, TTF, and TAM are moderated by consumers’ demographic characteristics warrants further examination.

In addressing these issues, this study aimed to investigate the factors affecting consumers’ m-health acceptance by proposing a comprehensive model based on SDT, TAM, and TTF, and to examine how the relationships differ across demographic characteristics. In the proposed model, SDT was used to capture user’s self-motivation for m-health acceptance, TAM represented users’ perceptions regarding the use of m-health, and TTF was used to capture user’s interaction experience with m-health, which was how the functionality of m-health affected their task performance. The following sections review the theoretical background and provide a detailed description of the development of the research hypotheses.

## 2. Theoretical Background and Research Hypotheses

### 2.1. Technology Acceptance Model

TAM, proposed by Davis [39], is one of the most widely used theories for user acceptance of technology [3]. The core idea of TAM is that individuals’ actual technology usage behaviors could be predicted by the behavioral intention to use the technology, which in turn is determined by two key constructs: perceived ease of use and perceived usefulness [39]. Moreover, the perceived ease of use has a direct impact on perceived usefulness. Both usage behaviors and behavioral intention are now commonly considered agents of technology acceptance. TAM has been widely used as the theoretical basis to explain the variance in technology usage and acceptance behaviors in such domains as e-learning [37], autonomous vehicles [40], and smart health [5]. A recent review showed that TAM could represent a good ground theory to examine the factors that affect the user acceptance of consumer-oriented health information technologies [3]. Due to its transferability to various contexts and predictive power over a wide range of information systems [41], we thus chose TAM as the central theoretical framework.

Referring to Davis [39], perceived usefulness in this study refers to the degree to which a person believes that using m-health would enhance his or her ability to manage their healthcare, while the perceived ease of use refers to the degree to which a person believes that using m-health would be free of effort. Previous studies have consistently demonstrated that perceived ease of use has a direct impact on perceived usefulness and that the perceived ease of use and perceived usefulness influence technology usage behaviors [3]. Thus, we hypothesized that:

**H1:** 
*Perceived usefulness positively affects the usage behaviors of m-health services.*


**H2:** 
*Perceived ease of use positively affects the usage behaviors of m-health services.*


**H3:** 
*Perceived ease of use positively affects the perceived usefulness of m-health services.*


### 2.2. Task–Technology Fit Model

Based on Goodhue and Thompson [32], task–technology fit is the extent to which technologies assist individuals in performing tasks. TTF suggests that a better fit between task and technology could improve efficiency and effectiveness when using the technology [32]. In the m-health context, a better task–technology fit would be achieved when m-health’s characteristics (e.g., real-time feedback, personalized healthcare services, and friendly interface) meet the requirements of individuals’ healthcare tasks, and vice versa. Previous studies have made endeavors to link TTF with technology acceptance and usage behaviors [33,41,42,43]. For example, Wu et al. investigated users’ continual intention to use massive open online courses (MOOCs) by integrating TTF and other acceptance models [41] and found that TTF was a significant predictor of perceived usefulness and perceived ease of use, which subsequently affected users’ acceptance of MOOCs. The antecedent role of TTF in shaping technology acceptance through the mediating role of perceived usefulness and perceived ease of use has also been confirmed in consumers’ use of smart home technology [33] and wearable healthcare devices [24], in students’ use of smartwatches for learning activities [33], and in college students’ use of e-learning in the era of COVID-19 [44]. In the m-health context, it is likely that only when users feel a good match between m-health functions and their healthcare tasks will they consider that the use of m-health can improve their performance during healthcare tasks. Similarly, as m-health could deliver healthcare services anytime and anywhere at a low cost, overcoming geographical, temporal, and even organizational barriers, users can complete their healthcare tasks more quickly and effortlessly [24]. Based on these arguments, we proposed that:

**H4:** 
*Task–technology fit positively affects the perceived ease of use of m-health services.*


**H5:** 
*Task–technology fit positively affects the perceived usefulness of m-health services.*


### 2.3. Self-Determination Theory of Motivation

SDT is a theory that emphasizes the importance of individuals’ self-motivation for the self-regulation of behaviors [45]. It posits that one’s self-motivation could be represented by three basic psychological needs: autonomy, competency, and relatedness, which in turn can affect individuals’ behaviors [46]. Previous studies have demonstrated the important roles of SDT constructs in technology acceptance are largely through the mediating effects of the perceived ease of use. For example, Fathali and Okada extended TAM with SDT to explore the factors affecting Japanese learners’ use of learning technology for out-of-class language learning and found that the three SDT constructs could yield significant impacts on the perceived ease of use and perceived usefulness [47]. Similarly, Nikou and Economides proposed a combined model of SDT and TAM to explain users’ acceptance of a mobile-based assessment [48], and they found that SDT constructs are more likely to affect acceptance through the mediating role of perceived ease of use. 

#### 2.3.1. Autonomy

Autonomy refers to users’ sense of regulation and self-control of their behaviors [27]. In this study, autonomy is defined as the degree to which users believe that they can use m-health for healthcare activities autonomously. Previous research has shown that autonomy had a direct impact on the perceived ease of use of mobile apps [48]. In addition, autonomy has been found to be related to the acceptance of e-learning systems [49]. It appears that if users feel autonomous when using m-health, they will feel m-health is easier to use and are more likely to use it in their healthcare activities. Therefore, we hypothesized that:

**H6a:** 
*Autonomy positively affects the perceived ease of use of m-health services.*


**H7a:** 
*Autonomy positively affects the usage behaviors of m-health services.*


#### 2.3.2. Relatedness

Relatedness refers to the desire to feel connected to and interact with others [46]. In this study, relatedness refers to how users perceive that they are connected with relevant stakeholders (e.g., peers or healthcare providers) when using m-health. Relatedness would be necessary for the use of m-health, as it can make people feel supported, cared for, and encouraged by other social members in their healthcare activities. Previous studies suggest that relatedness could positively influence perceived playfulness, a similar construct to perceived ease of use [50]. Likewise, Khan found that relatedness had a positive effect on the usage behaviors of MOOCs [51]. In this regard, we proposed that the increase in relatedness is related to higher levels of perceived ease of use and usage behaviors. Therefore, we hypothesized that:

**H6b:** 
*Relatedness positively affects the perceived ease of use of m-health services.*


**H7b:** 
*Relatedness positively affects the usage behaviors of m-health services.*


#### 2.3.3. Competence

Competence refers to users’ desire to be effective and efficient when performing certain tasks [48]. In the m-health context, a sense of competence would be developed when users believe that they are skillful at using m-health. In fact, the use of m-health requires users to find, understand, and respond to professional health information and services, in addition to the ability to operate complex mobile applications. Successful completion of healthcare activities with m-health can make users feel more competent and confident in using m-health, thereby increasing its perceived ease of use. Previous studies have shown that a higher level of competence would result in a higher level of perceived ease of use, and facilitate the user acceptance of e-learning tools [48,50]. Therefore, we hypothesized that:

**H6c:** 
*Competence positively affects the perceived ease of use of m-health services.*


**H7c:** 
*Competence positively affects the usage behaviors of m-health services.*


In summary, an m-health acceptance model, depicted in Figure 1, was developed based on the above-mentioned hypotheses. In the model, autonomy, relatedness, and competence were grouped to represent users’ self-motivation, TAM represented users’ perceptions regarding the use of m-health, and TTF was used to indicate the technical aspects of users’ interaction experiences with m-health.

## 3. Methods

### 3.1. Participants 

Data were collected using the Wenjuanxing Survey (www.sojump.com, accessed on 20 December 2020), one of the most popular online survey platforms in China, which has been widely used in previous studies. The survey platform owns a sample database of 2.6 million active members with a wide range of demographic characteristics and geographical distributions in China. The target population comprised adults with at least six months of m-health usage experience, as they were familiar with m-health and could fill out the questionnaire well. To reduce possible biased reactions and ethical and privacy considerations, participants were informed that they would be asked about their demographics and their views and perceptions of m-health, that the questionnaire would be answered anonymously, that the data would be de-identified, and that they should answer questionnaire items seriously and honestly. The study was approved by the Institutional Review Board of Shenzhen University. Informed consent was obtained from the participants when they agreed to complete the survey. The survey platform randomly distributed the questionnaire to 700 eligible participants in their sample database, of which, 623 valid samples were received (response rate: 89%) and used for data analysis. The descriptive data of the participants are summarized in Table 2.

### 3.2. Instruments 

The questionnaire was created based on an extensive review of relevant studies and adapted from validated measurement scales. It was further refined based on consultation with three experts in questionnaire design and m-health to improve its clarity and readability. Some measurement items were modified as necessary to reflect the m-health context and better fit our study scenario. The questionnaire consisted of two sections. The first section gave descriptions of m-health that could facilitate participants’ understanding of the m-health context to be surveyed. A brief description of m-health was provided as follows, “M-health services refer to healthcare services that are delivered through mobile technology (e.g., smartphones, tablets, and wearable devices). Representative examples of widely used m-health applications included Chunyu Doctor, Dingdang Medicine Express, and Ping An Good Doctor”. The second section listed items that were used to collect participants’ demographic information and responses to the proposed constructs. The measurement items and their sources are presented in Appendix A. All the items were measured with a 5-point Likert-type scale ranging from 1 (strongly disagree) to 5 (strongly agree). 

### 3.3. Data Analysis

A two-step structural equation modeling (SEM) approach was performed to verify the measurement model and the structural model. The first step was to assess the reliability, convergent validity, and discriminant validity of the measurement constructs. The reliability is considered good if Cronbach’s alpha is larger than 0.70, and it is acceptable if Cronbach’s alpha is larger than 0.6 [52]. Convergent validity is considered acceptable if the composite reliability of the construct is higher than 0.70 and the average variance extracted (AVE) of the construct is larger than 0.50 [53]. Discriminant validity is verified if the square root of the AVE for a given construct is larger than its correlations with any other constructs [53].

The second step was to check the structural model by assessing path coefficients and model fit. The hypothesized structural model and the data were tested by SEM, and each of the path coefficients was estimated using the maximum likelihood estimates method. Seven commonly used goodness-of-fit indices were employed to assess the overall model fit. A good fit was indicated by the ratio of chi-square to degrees of freedom (χ^2^/df < 3), the goodness-of-fit index (GFI ≥ 0.90), the adjusted goodness-of-fit index (AGFI ≥ 0.80), the comparative fit index (CFI ≥ 0.90), the incremental fit index (IFI ≥ 0.90), the Tucker Lewis index (TLI ≥ 0.90), and root mean square error of approximation (RMSEA < 0.08) [54,55]. 

In addition, we used multi-group SEM analysis to investigate whether the path coefficients of the hypothesized relationships were equivalent across gender (male vs. female), age (younger vs. middle-aged adults, with median = 30 years as the cut-off point), and usage experience (less experienced and more experienced users, with three years as the cut-off point). The data analyses were conducted using AMOS 24. 

## 4. Results

### 4.1. Reliability and Validity Assessment 

Table 3 presents the results on factor loading, Cronbach’s α, composite reliability, and AVE. The Cronbach’s α value was larger than 0.70 for relatedness, competence, TTF, perceived ease of use, perceived usefulness, and usage behaviors, whereas the value was only slightly below 0.70 for autonomy (α = 0.66). Thus, the constructs were considered to have good internal consistency. The value of composite reliability for all the constructs varied from 0.82 to 0.91. The value of AVE for all the constructs ranged from 0.54 to 0.78. Therefore, convergent validity for the constructs was achieved. Discriminant validity was satisfactory as the square root of the AVE for each construct was larger than its correlations with any other constructs (Table 4). 

### 4.2. Model Testing

The results of the model fit are summarized in Table 5. The results showed that the proposed model had a good model fit. Figure 2 depicts the results of the estimated model and Table 6 summarizes the results of hypotheses testing. Overall, the perceived usefulness, perceived ease of use, and three STD constructs explained 81.1% of the variance in usage behaviors. The perceived ease of use and task–technology fit explained 77.4% of the variance in perceived usefulness, while the three STD constructs and task–technology fit accounted for 39.5% of the variance in the perceived ease of use. 

The perceived usefulness (β = 0.619, *p* < 0.001) and perceived ease of use (β = 0.243, *p* < 0.01) were found to positively affect usage behaviors. Therefore, H1 and H2 were supported. H3 was also supported, as the perceived ease of use showed a positive effect on the perceived usefulness (β = 0.319, *p* < 0.001). Task–technology fit had a positive influence on perceived usefulness (β = 0.657, *p* < 0.001), supporting H5. Two STD factors (i.e., relatedness (β = 0.137, *p* < 0.05) and competence (β = 0.130, *p* < 0.05)) had a positive influence on the perceived ease of use, supporting H6b and H6c. The results failed to provide support for other hypotheses.

### 4.3. Multi-Group SEM Analysis

The results showed that most of the path coefficients in the proposed model remained equal across gender, age, and usage experience groups (Table 7). The gender groups demonstrated significant differences in the two paths. More specifically, perceived usefulness was more strongly related to usage behaviors for males compared with females (Z score = −1.978, *p* < 0.05). However, autonomy was more strongly related to usage behaviors for the female group than for the male group (Z score = 2.120, *p* < 0.05). There was no significant difference in path coefficients across age and usage experience. 

## 5. Discussion

Regarding the overall model, the results supported the innate psychological needs of m-health users in terms of relatedness and competence as the significant self-initiated motivators of the perceived ease of use. Referring to SDT [27], the results indicated that if users could connect to, interact with, and care for relevant stakeholders (i.e., doctors, nurses, peers, and their family members) or m-health activities, and feel more effective and capable in using m-health services, they were more likely to perceive m-health as easy to use. Similar to previous studies [48,50], the results showed that autonomy was not significantly related to perceived ease of use. This is in line with the viewpoints of various cross-cultural researchers that autonomous activities were not likely to be strongly valued in Eastern cultures [56]. In contrast, the experience of being encouraged by relevant important others and competence leading to success might have a stronger impact on people’s behavioral performance [57]. The results from this study call further attention to the cultural differences facilitating users’ m-health usage behaviors. 

Moreover, contrary to previous studies [48], the results showed that all three self-motivations from SDT were not significantly related to users’ perceived usefulness. Some previous studies revealed that intrinsic or self-motivations were more likely to lead to sophisticated practices rather than superficial engagement [58]. The results again provided possible explanations for the significant relationships between intrinsic motivations and the perceived ease of use, suggesting intrinsic motivations were more likely to be related to experience-specific cognitive factors rather than utility-related cognitive factors. In terms of the critical antecedents of perceived usefulness, the results of our study suggested a significant correlation between task–technology fit and perceived usefulness. This finding is consistent with many previous studies on various digital technologies, such as wearable healthcare devices [24], smartwatches [33], and online learning [37]. The finding should be reasonable, as a task–technology fit is widely recognized as a factor affecting user performance in technology utilization [32]. 

Our research also confirmed the findings from previous technology acceptance studies in the field of health informatics [24,59]. More specifically, our study found that the perceived ease of use has a strong positive impact on perceived usefulness and that the perceived ease of use and perceived usefulness collectively demonstrate strong significant effects on usage behaviors of m-health. These findings indicated that when users perceived using m-health as effortless, they were more likely to perceive using m-health as efficient and effective, and these two perceptions, in turn, were likely to result in more actual usage of m-health. Consistent with many previous studies [38,47], the findings verified the more important role of perceived usefulness in facilitating usage behaviors compared with the perceived ease of use, indicating that users were concerned more about usefulness when deciding to use m-health services. 

The multi-group analyses revealed that two relationships between the determinants (i.e., perceived usefulness and autonomy) and usage behaviors were significantly moderated by gender. In particular, although the relationships between autonomy and usage behaviors were not significant for either male or female groups, a significant path difference was found between the two gender groups. This appears to suggest that autonomy might work differently for males and females. While autonomy tended to decrease males’ willingness to use m-health services, it has a very limited impact on the usage behaviors of females. In addition, the results showed that perceived usefulness was much more strongly related to usage behaviors in males than in females. The results imply that when males and females have a similar perceived usefulness of m-health, males are much more likely to demonstrate an actual usage of m-health. The results are consistent with many previous studies which revealed that males valued perceived usefulness more than females in their decision-making to accept a technology [37]. 

Although the relationships were not significantly moderated by age and usage experience, some relationships demonstrated great differences in their path coefficients, similar to previous studies [37]. It should be noted that relatedness and perceived ease of use were only significantly related to the usage behaviors of middle-aged users. It might be possible that middle-aged users had more desire for social connection and were less familiar with advanced technologies than younger users. These findings suggested that middle-aged people valued connection, interaction with others, and ease of use when using m-health services. 

For the experienced group, it should be highlighted that relatedness was only significantly related to the perceived ease of use, and perceived usefulness was only significantly related to usage behaviors for the less experienced group. Given the crucial roles of perceptions, these results implied that less experienced users highly valued their interaction and connection when using m-health and the utility of m-health services. The results align with previous studies [3,37] and indicate that social influence-related external motivation and perceived usefulness are critical to facilitating users’ initial acceptance and usage of technology.

## 6. Conclusions

The successful implementation of m-health services depends largely on user acceptance. Yet, the prominent motivating factors considering self-motivations and the task–technology fit for using such services warrant further examination in the field of human-computer interaction. Moreover, the acceptance behaviors of m-health services are relatively personalized and social processes which may differ across users with different social demographic backgrounds. However, how users’ acceptance behaviors are moderated by important social demographic backgrounds has not yet been fully understood. Based on these gaps, the current study proposed an m-health acceptance theoretical model combining SDT, TAM, and TTF to test users’ acceptance of m-health, and explored the moderating effects of users’ gender, age, and usage experience on the modeled relationships. The results of SEM analyses on 623 users of m-health services validated the proposed integrated model, with the variance explained in the usage behavior of m-health being more than 75% in the overall and the multi-group models. The implications of the present study were discussed, and the limitations and future research possibilities were presented. 

### 6.1. Implications

The theoretical contributions of this study were three-fold. First, this study, with experienced Chinese users, shows the applicability of an integrated model of SDT, TTF, and TAM for explaining the user acceptance and usage behaviors of m-health. This allowed the integrated m-health acceptance model to explain more of the variance in user acceptance and usage behaviors compared with many previous studies [5,11,38]. The findings implied that the integration of SDT and TTF with TAM is more effective in examining consumers’ acceptance of m-health. Second, our integrated m-health acceptance model not only helped us understand usage behaviors from the perspectives of motivational needs and perceptions regarding m-health but also elaborated the role of self-motivation in shaping users’ intentional use of m-health. Third, the findings demonstrated the importance of adding task–technology fit components into acceptance theories in exploring the important antecedents of acceptance and usage behaviors of m-health and other health information technologies. 

The findings also provided practical suggestions for practitioners to better design and promote m-health use for healthcare activities across consumers with different personal backgrounds. Apart from the conventional recognition of the important role of usefulness and ease of use in the acceptance of m-health services, our study further revealed specific strategies to improve the usefulness and ease of use of the technology. In particular, m-health developers should attach great importance to consumer self-motivation factors so that these services can meet consumers’ motivational needs for relatedness and competence. For example, designers can develop social functions that enable consumers to communicate with healthcare providers and peers in real-time to meet relatedness requirements so that consumers can feel socially connected and supported. Competence can also be satisfied by strengthening their knowledge and skills for operating such complex m-health applications and by empowering them to take control of their health and healthcare process. However, it should be noted that, unlike many other types of technologies [48], autonomy appears to be less important in determining consumers’ acceptance and usage behaviors for m-health. Thus, designers should not emphasize autonomous use for m-health. In addition, as competence and relatedness exerted their effects through the mediating role of perceived ease of use, designers must guarantee that consumers can satisfy these requirements easily when interacting with m-health.

In addition, our results imply that the usefulness of m-health can be reinforced if the technology achieves a good task–technology fit in service provision. When the characteristics of m-health meet the requirements of consumers’ health management tasks, m-health can be considered more useful. Therefore, designers should ensure that m-health is able to meet consumers’ healthcare requirements by matching its characteristics and functions with health management tasks. This can be achieved by measures to ensure the “anywhere” and “anytime” features that m-health provides (e.g., real-time data collection, quick feedback to health inquiry, and compatibility with a diverse wide of healthcare activities).

Finally, although most of our examined relationships appear robust across gender, age, and usage experience, practitioners should be aware of the differences in the use of m-health among consumers with different demographic backgrounds and take corresponding measures to cope with the differences. For example, perceived usefulness was much more strongly related to usage behaviors for males than females. It is thus suggested that utility and efficiency could be more emphasized for male users in m-health functions. Relatedness was only significantly related to the usage behaviors of middle-aged users. This suggests that connection and communication functions could be promoted among middle-aged users instead of younger adults. Overall, it is recommended that designers should prioritize their resources in the design and implementation of m-health for different groups based on these findings so as to maximize the acceptance and effectiveness of m-health services.

### 6.2. Limitations and Future Work

This study has several limitations which could guide future research directions. First, while a large proportion of older people have an urgent need for efficient and convenient healthcare services [11], the generalization of our research findings to older people should be cautious, as most of our participants were younger and middle-aged users. Therefore, future research is encouraged to explore how older users perceive, accept, and use these complex m-health services. Second, our study failed to focus on a specific m-health service but examined general m-health services. However, user acceptance might vary in different m-health applications as the healthcare activities and functions enabled by applications are different. How healthcare activities and m-health functions affect users' acceptance deserves further exploration. Finally, like many previous studies [3,5,24,38], our study only examined user acceptance in a cross-sectional study. However, the continuous acceptance of m-health services is the key to the long-term success of m-health applications. Thus, future studies could consider extending our research in a longitudinal way to examine the changes in user acceptance and usage behaviors of m-health. 

## Figures and Tables

**Figure 1 healthcare-11-01550-f001:**
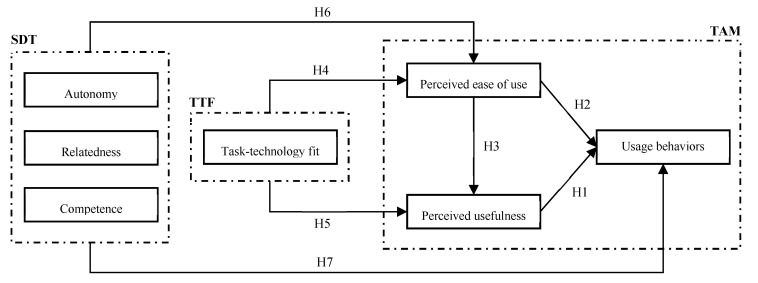
The proposed m-health acceptance model.

**Figure 2 healthcare-11-01550-f002:**
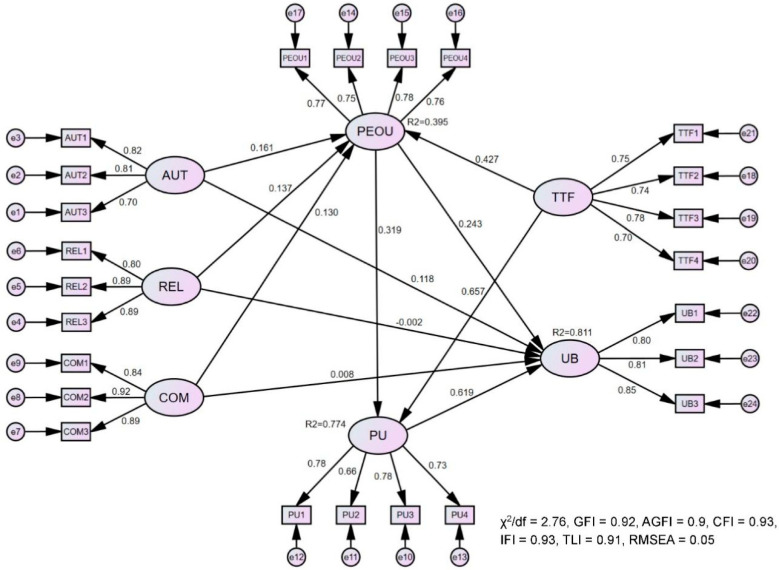
Structural model evaluation results.

**Table 1 healthcare-11-01550-t001:** A summary of m-health acceptance studies and their major findings.

Study	Country	Type of m-Health	Sample Size	Theory Base	Factors Examined in the Model (Significant Factors Are Emboldened)
Zhang et al., 2014 [12]	China	M-health service	481	TRA	**ATT**, **FC**, **SN**, and **gender**
Deng et al., 2014 [11]	China	M-health service	424	TAM, TPB, and value–attitude–behavior model	**ATT**, **PBC**, **perceived value**, RC, SN, TA, self-actualization need, perceived physical condition, and **age**
Gao et al., 2015 [13]	China	Wearable healthcare technology	462	UTAUT2, protection motivation theory, and privacy calculus theory	**PEOU**, **PU**, **PPR**, **SI**, **self-efficacy**, **perceived severity**, and **perceived vulnerability**
Cho, 2016 [14]	Korea	Mobile health Apps	343	TAM	**PEOU**, **PU**, and **confirmation**
Hoque and Sorwar, 2017 [15]	Bangladesh	M-health services	274	UTAUT	**PEOU**, **PU**, **SI**, **RC**, **TA**, and FC
Zhu et al., 2017 [16]	China	Mobile chronic disease management systems	279	TAM	**PEOU**, **PU**, **TA**, **perceived disease threat**, **initial trust**, and perceived risk
Zhang et al., 2017 [17]	China	M-health services	650	TAM	**PEOU**, **PU**, **self-efficacy**, and **response-efficacy**
Cilliers et al., 2018 [18]	South Africa	Mobile phone-based health information	202	UTAUT	**PU**, **ATT**, **SI**, and **mobile experience**
Alaiad et al., 2019 [20]	Jordan	M-health services	280	UTAUT, dual-factor model, and health belief model	**PEOU**, **PU**, **SI**, ****RC**, perceived health threat**, **m-health app quality**, **life quality expectancy**, **security**, and **privacy risks**
Nunes and Castro, 2019 [19]	Portugal	Mobile health applications	394	UTAUT	PEOU, **PU**, SI, FC, **age**, **smartphone experience**, and gender
Liu and Tao, 2022 [5]	China	Smart healthcare services	769	TAM	**PEOU**, **PU**, **trust**, **personalization**, **loss of privacy**, and **anthropomorphism**
Wang et al., 2022 [6]	China	Mobile medical platforms	389	TAM, TPB	**PEOU**, **PU**, **ATT**, **PBC**, **PPR**, SI, **perceived convenience**, and **perceived credibility**
Alsyouf et al., 2022 [28]	Saudi Arabia	Exposure detection apps	586	TAM	**PEOU**, **PU**, **perceived privacy**, and social media awareness
Cao et al., 2022 [29]	China	M-health Apps	500	Digital content value chain framework	**User–functional interaction**, **user–information interaction**, **user–doctor interaction**, **healthcare assurance capacity**, **healthcare confidence**, and **parasocial relationships**
Chuenyindee et al., 2022 [30]	Thailand	COVID-19 contact-tracing application	800	TAM and protection motivation theory	**PEOU** and **PU**
Lu et al., 2023 [21]	China	Mobile medical consultation	475	Information systems continuance model	**Immediacy**, **telepresence**, **intimacy**, **substitutability**, **pandemic-induced anxiety**, and **satisfaction**
Alsyouf et al., 2023 [31]	Saudi Arabia	Personal health record system	389	TAM	**PEOU**, **PU**, **security**, privacy, and usability

Note: TRA, Theory of Reasoned Action; TPB, Theory of Planned Behavior; UTAUT, The Unified Theory of Acceptance and Use of Technology; ATT, attitude; PEOU, perceived ease of use; PU, perceived usefulness; FC, facilitating conditions; PPR, perceived privacy risk; PBC, perceived behavioral control; SI, social influence; SN, subjective norm; RC, resistance to change; and TA, technology anxiety.

**Table 2 healthcare-11-01550-t002:** Demographic characteristics of the participants.

Items	Classification	Number of Participants	Percentage (%)
Gender	Male	266	42.7%
	Female	357	57.3%
Age	18–29	302	48.5%
	30 or above	321	51.5%
Education	High school or less	23	3.7%
	University/college	547	87.8%
	Postgraduate	53	8.5%
Duration of using m-health	Less than 3 years	403	64.7%
	3 years or more	220	35.3%
Frequency of using m-health	Yearly	352	56.5%
	Monthly	207	33.2%
	Weekly	56	9%
	Daily	8	1.3%
Main purpose of using m-health	Online healthcare consultation	302	48.5%
	Appointment registration	83	13.3%
	Medical/health information inquiry	112	18%
	Self-monitoring of health status	70	11.2%
	Purchase of medication	10	1.6%
	Comprehensive health management	46	7.4%

**Table 3 healthcare-11-01550-t003:** The factor loading, Cronbach’s α, composite reliability, and AVE results for the constructs.

Constructs	Items	Factor Loadings	Cronbach’s α	Composite Reliability	Average Variance Extracted (AVE)
Autonomy	AUT1	0.82	0.66	0.82	0.61
	AUT2	0.81			
	AUT3	0.70			
Relatedness	REL1	0.80	0.82	0.89	0.74
	REL2	0.89			
	REL3	0.89			
Competence	COM1	0.84	0.86	0.91	0.78
	COM2	0.92			
	COM3	0.89			
Task–technology fit	TTF1	0.75	0.72	0.83	0.55
	TTF2	0.74			
	TTF3	0.78			
	TTF4	0.70			
Perceived usefulness	PU1	0.78	0.72	0.83	0.55
	PU2	0.66			
	PU3	0.78			
	PU4	0.73			
Perceived ease of use	PEOU1	0.77	0.76	0.85	0.59
	PEOU2	0.75			
	PEOU3	0.78			
	PEOU4	0.76			
Usage behaviors	UB1	0.80	0.75	0.86	0.67
	UB2	0.81			
	UB3	0.85			

**Table 4 healthcare-11-01550-t004:** Discriminant validity.

	AUT	REL	COM	TTF	PEOU	PU	UB
AUT	0.78						
REL	0.42 **	0.86					
COM	0.36 **	0.35 **	0.88				
TTF	0.66 **	0.42 **	0.40 **	0.74			
PEOU	0.46 **	0.34 **	0.32 **	0.51 **	0.77		
PU	0.61 **	0.42 **	0.34 **	0.63 **	0.54 **	0.74	
UB	0.53 **	0.31 **	0.29 **	0.51 **	0.54 **	0.62 **	0.82

UB, usage behaviors; PU, perceived usefulness; PEOU, perceived ease of use; AUT, autonomy; REL, relatedness; COM, competence; and TTF, task–technology fit. ** *p* < 0.01.

**Table 5 healthcare-11-01550-t005:** Measurement model fit.

Model Fit Indices	Recommended Value	Tested Model
χ^2^/df	<3.0	2.76
GFI	>0.9	0.92
AGFI	>0.8	0.90
CFI	>0.9	0.93
IFI	>0.9	0.93
TLI	>0.9	0.91
RMSEA	<0.08	0.05

**Table 6 healthcare-11-01550-t006:** Hypotheses testing results.

Hypotheses	Path	Path Coefficient (β)	Standard Deviation	t Value	*p* Value	Supported? (Yes/No)
H1	PU→UB	0.619 ***	4.143	4.507	<0.001	Yes
H2	PEOU→UB	0.243 **	2.221	2.973	0.003	Yes
H3	PEOU→PU	0.319 ***	1.348	5.267	<0.001	Yes
H4	TTF→PEOU	0.427	6.889	1.566	0.117	No
H5	TTF→PU	0.657 ***	1.647	9.050	<0.001	Yes
H6a	AUT→PEOU	0.161	9.385	0.579	0.562	No
H6b	REL→PEOU	0.137 *	0.899	2.195	0.028	Yes
H6c	COM→PEOU	0.130 *	0.824	2.418	0.016	Yes
H7a	AUT→UB	0.118	3.894	1.109	0.267	No
H7b	REL→UB	−0.022	0.749	−0.474	0.636	No
H7c	COM→UB	0.008	0.749	0.170	0.865	No

UB, usage behaviors; PU, perceived usefulness; PEOU, perceived ease of use; AUT, autonomy; REL, relatedness; COM, competence; and TTF, task–technology fit. * *p* < 0.05, ** *p* < 0.01, and *** *p* < 0.001.

**Table 7 healthcare-11-01550-t007:** Multi-group analysis by gender, age, and usage experience.

Hypotheses	Whole		Gender		Age			Usage Experience		
		Male (N = 266)	Female (N = 357)	Z Score	Younger Users (N = 302)	Middle-Aged Users (N = 321)	Z Score	Less Experienced Users (N = 403)	More Experienced Users (N = 220)	Z Score
H1: PU→UB	0.619 ***	0.745 **	0.382 *	−1.978 *	0.591 *	0.565 *	−0.032	0.614 ***	0.4	−0.544
H2: PEOU→UB	0.243 **	−0.046	0.326 *	1.707	0.291	0.28 *	−0.470	0.24 **	0.368 *	0.566
H3: PEOU→PU	0.319 ***	0.343 ***	0.288 ***	−0.206	0.419 ***	0.292 ***	−1.777	0.281 ***	0.46 ***	0.960
H4: TTF→PEOU	0.427	0.564	0.31	−0.639	0.516	0.277	−0.187	0.819	0.221	−0.775
H5: TTF→PU	0.657 ***	0.68 ***	0.688 ***	−0.228	0.561 ***	0.691 ***	0.997	0.663 ***	0.561 ***	−0.655
H6a: AUT→PEOU	0.161	0.056	0.271	0.415	0.13	0.273	0.297	−0.28	0.468	0.985
H6b: REL→PEOU	0.137 *	0.1	0.188 **	0.405	0.163	0.127	0.113	0.197 *	0.048	−1.199
H6c: COM→PEOU	0.13 *	0.147 *	0.076	−0.953	0.15	0.099	−0.198	0.126	0.117	−0.162
H7a: AUT→UB	0.118	−0.705	0.214	2.120 *	0.091	0.174	0.294	0.15	0.177	0.040
H7b: REL→UB	−0.022	0.091	−0.023	−0.951	0.037	0.134 *	−1.831	−0.033	−0.027	0.100
H7c: COM→UB	0.008	−0.014	0.037	0.518	−0.012	−0.011	0.013	−0.024	0.042	0.673
R^2^ (overall model)	81.1%	83.2%	86.4%		83.5%	83.7%		82.1%	76.5%	

UB, usage behaviors; PU, perceived usefulness; PEOU, perceived ease of use; AUT, autonomy; REL, relatedness; COM, competence; and TTF, task–technology fit. * *p* < 0.05, ** *p* < 0.01, and *** *p* < 0.001.

## Data Availability

The data presented in this study are available on request from the corresponding author. The data are not publicly available due to privacy issue.

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
