# Peer review of "Modeling Consumer Acceptance and Usage Behaviors of m-Health: An Integrated Model of Self-Determination Theory, Task–Technology Fit, and the Technology Acceptance Model"

_healthcare, 2023, doi:10.3390/healthcare11111550_

Round 1
Reviewer 1 Report
Thank you very much for the opportunity to review the paper titled “Modelling consumer acceptance and usage behaviours of m-health: An integrated model of Self-Determination Theory, Task-Technology Fir and Technology Acceptance Model”.
As it is correctly mentioned in the manuscript, there are numerous studies that explored the acceptance and adoption of m-health, so it is important to justify why this research is needed. One way to do that will be to provide a brief summary of what other studies did and how the findings of this paper differ. Just mentioning that this study will employ SDT is not a strong enough reason (why constructs in SDT may be effective in developing an understanding of consumers’ usage behaviours in relation to m-health?).
Instead of the provided literature review on the selected theories, it might be better to provide a review on m-health research. This approach will help justify the need to conduct this study.
The need to combine the selected 3 theories should be better explained. When discussing the combination of TTF and TAM, the following study by Marikyan et al. (2021) might be useful. Also, a bit more justification is required to conclude that TTF can be an antecedent of PU and PEU.
The reliability of autonomy is 0.66, but above it is mentioned that Cronbach’s Alfas are higher than 0.70.
The theoretical contribution(s) of this study should be better justified. For the time being, it is difficult to see as to why combining SDT, TTF and TAM can be considered a theoretical combination.
Reference list:
Marikyan, D., Papagiannidis, S. and Alamanos, E., 2021. “Smart home sweet smart home”: An examination of smart home acceptance. International Journal of E-Business Research (IJEBR), 17(2), pp.1-23.
Author Response
Response to Reviewer #1:
Thank you very much for the opportunity to review the paper titled “Modelling consumer acceptance and usage behaviours of m-health: An integrated model of Self-Determination Theory, Task-Technology Fir and Technology Acceptance Model”.
Comment #1
As it is correctly mentioned in the manuscript, there are numerous studies that explored the acceptance and adoption of m-health, so it is important to justify why this research is needed. One way to do that will be to provide a brief summary of what other studies did and how the findings of this paper differ. Just mentioning that this study will employ SDT is not a strong enough reason (why constructs in SDT may be effective in developing an understanding of consumers’ usage behaviours in relation to m-health?).
Response:
The authors appreciate the comment made by the reviewer. As suggested by the reviewer, we have provided a brief summary of previous studies on m-health acceptance, as shown in the new Table 1 (Page 2), and identified research limitations of previous studies. In addition, the incorporation of SDT in our research model has been well justified accordingly. In particular, why constructs in SDT could be effective in developing an understanding of consumers’ usage behaviours in relation to m-health has been well explained, as shown below.
“Although existing studies have examined a set of variables to facilitate m-health acceptance, such as perceptions of m-health (e.g., perceived ease of use, perceived usefulness and perceived privacy risk), social factors (e.g., subjective norms and social influence) and personal factors (e.g., age, gender and experience) (See Table 1 for a summary), there is a surprising dearth of research on motivational and technical factors that are crucial in understanding consumers’ usage of m-health. First, unlike many other health information technologies that were mainly used by health professionals, m-health is mainly used by younger and middle-aged consumers, who are well educated and skillful at complicated technology operations, and thus tend to dominate the interaction process with m-health. This means that the use of m-health is largely characterized by the following features: self-domination, employment of technical skills, and frequent interaction with online peers or healthcare providers [26]. The three features are well matched with three basic self-motivation constructs of Self-Determination Theory (SDT) of Motivation (i.e., autonomy, competence, and relatedness)[27]. The feature of self-domination indicates implications for autonomous use of m-health to deal with healthcare issues, the feature for employment of technical skills could be considered as the competent use of m-health, while the frequent interaction feature represents the relatedness requirement in SDT. Therefore, the three self-motivation constructs in SDT may be effective to understand consumers’ usage behaviors of m-health.” (Page 2, lines 70-89)
Comment #2
Instead of the provided literature review on the selected theories, it might be better to provide a review on m-health research. This approach will help justify the need to conduct this study.
Response:
The authors appreciate the comment made by the reviewer. As suggested by the reviewer, we have provided a brief summary of previous studies on m-health acceptance, as shown in the new Table 1 (Page 2), and identified research limitations of previous studies. Specifically, we argue that there is a surprising dearth of research on motivational and technical factors that are crucial in understanding consumers’ usage of m-health. This leads to our motives to conduct this study.
To recap, first, “the troubling problem of non-acceptance or under-use continues to be an essential concern in m-health practice.”(Page 2, lines 62-63)
Second, we argue that “Although existing studies have examined a set of variables to facilitate m-health acceptance, such as perceptions of m-health (e.g., perceived ease of use, perceived usefulness and perceived privacy risk), social factors (e.g., subjective norms and social influence) and personal factors (e.g., age, gender and experience) (See Table 1 for a summary), there is a surprising dearth of research on motivational and technical factors that are crucial in understanding consumers’ usage of m-health.”(Page 2, line 70-76). This leads to our motives to conduct this study.
Finally, we have well explained and justified the integration of TAM, STD and TTF to examine consumers’ m-health acceptance.
“Although TTF and TAM have been previously used to explain acceptance in a variety of health informatics applications, their underlying theoretical assumptions are different. While TTF highlights the fit between technology characteristics (e.g., the functionality of m-health) and task requirements [28], TAM is concerned with users’ perceptions (e.g., beliefs and attitudes) about using the technology or service [31]. The results from previous studies have suggested that the combination of TAM and TTF better explains the variances in technology acceptance than TAM and TTF alone [24]. However, few studies have examined consumers’ m-health acceptance from integrated perspectives of technology perceptions, users’ self-motivations, and task-technology fit.” (Page 3, lines 109-117)
Comment #3
The need to combine the selected 3 theories should be better explained. When discussing the combination of TTF and TAM, the following study by Marikyan et al. (2021) might be useful. Also, a bit more justification is required to conclude that TTF can be an antecedent of PU and PEU.
Response:
The authors appreciate the comment made by the reviewer. As suggested by the reviewer, we have reinforced our justification of why we use and combine the three theories (Please kindly see the last paragraph in Page 2 and the last two paragraphs in Page 3). To be specific, we have updated the rationale to incorporate SDT in our research model the last paragraph in Page 2. Then, we have well explained why TTF should be integrated in our model (the last two paragraphs in Page 3). Finally, the combination of the three theories has been explained, as shown below.
“Therefore, we inferred that whether users accept m-health or not may not only be determined by the abovementioned motivational factors, but also rely on a good task-technology fit. Although TTF and TAM have been previously used to explain acceptance in a variety of health informatics applications, their underlying theoretical assumptions are different. While TTF highlights the fit between technology characteristics (e.g., the functionality of m-health) and task requirements [28], TAM is concerned with users’ perceptions (e.g., beliefs and attitudes) about using the technology or service [31]. The results from previous studies have suggested that the combination of TAM and TTF better explains the variances in technology acceptance than TAM and TTF alone [24]. However, few studies have examined consumers’ m-health acceptance from integrated perspectives of technology perceptions, users’ self-motivations, and task-technology fit.” (Page 3, lines 107-117)
The hypothesis that TTF can be an antecedent of PU and PEU has been justified with more details.
“For example, Wu et al. investigated users’ continuance intention to use massive open online courses (MOOCs) by integrating TTF and other acceptance models [37], and found that TTF was a significant predictor of perceived usefulness and perceived ease of use, which subsequently affected users’ acceptance of MOOCs. More recently, Marikyan et al. also integrated the TTF and TAM to explore consumers’ smart home technology use, and the results supported the significant effect of TTF on both perceived ease of use and perceived usefulness [29]. The antecedent role of TTF in shaping technology acceptance through the mediating role of perceived usefulness and perceived ease of use has also been confirmed in consumers’ use of healthcare wearable devices [24], in students’ use of smartwatches for learning activities [29] and in college students’ use of e-learning in the era of COVID-19 [41,42]. In the m-health context, it is likely that only when users feel a good match between m-health functions and their healthcare tasks, they will consider that the use of m-health can facilitate their healthcare tasks and improve related task performance. Similarly, as the characteristics of m-health enable healthcare services to be delivered anytime and anywhere at low cost, overcoming geographical, temporal, and even organizational barriers, users will complete their health tasks more quickly and with less effort [24].” (Page 5, lines 181-197)
Marikyan et al.’s (2021) study can provide a good reference for this study and have been well cited in the abovementioned section.
References:
Marikyan, D., Papagiannidis, S. and Alamanos, E., 2021. “Smart home sweet smart home”: An examination of smart home acceptance. International Journal of E-Business Research, 17(2), pp.1-23.
Comment #4
The reliability of autonomy is 0.66, but above it is mentioned that Cronbach’s Alfas are higher than 0.70.
Response:
The authors appreciate the comment made by the reviewer. In the manuscript, the original statement is “All constructs had a Cronbach's α larger than 0.70, except for autonomy (Cronbach's α = 0.66).” (page 9, lines 305-306). We understand that the "all the ...., except for...." structure may confuse readers. We have revised the sentence as follows:
"The Cronbach's α value was larger than 0.70 for relatedness, competence, TTF, perceived ease of use, perceived usefulness and usage behaviors, whereas the value was only slightly below 0.70 for autonomy (α = 0.66)." (Page 8, lines 322-324)
Comment #5
The theoretical contribution(s) of this study should be better justified. For the time being, it is difficult to see as to why combining SDT, TTF and TAM can be considered a theoretical combination.
Response:
The authors appreciate the comment made by the reviewer. We have discussed the theoretical contribution of the current study on page 14, lines 435-446. To better present and clarify the theoretical contributions, we have revised this part as follows:
“The theoretical contributions were three-fold. First, this study, with experienced Chinese users, shows the applicability of an integrated model of SDT, TTF and TAM for explaining user acceptance and usage behaviors of m-health. This allowed the integrated m-health acceptance model to have explained more variance of user acceptance and usage behaviors compared with many of previous studies [5,11,34]. The findings implied that the integration of SDT and TTF with TAM is more effective in examining consumers’ acceptance of m-health. Second, our integrated m-health acceptance model not only helped understanding usage behaviors from the perspectives of motivational needs and perceptions regarding m-health, but also elaborated the role of self-motivation in shaping users’ intentional use of m-health. Third, the findings also demonstrated the importance of adding task-technology fit components into acceptance theories in exploring important antecedents of acceptance and usage behaviors of m-health and other health information technologies.” (Page 13, lines 451-463)

Reviewer 2 Report
Modelling consumer acceptance and usage behaviors of m- health: An integrated model of Self-Determination Theory, Task-Technology Fit and Technology Acceptance Model
Dear Authors,
It is possible to consider your paper entitled " Modelling consumer acceptance and usage behaviors of m- health: An integrated model of Self-Determination Theory, Task-Technology Fit and Technology Acceptance Model" for publication only after some revision. I have listed my comments below.
1. From a scientific perspective, the title is well-formulated.
2. The abstract is well organized and easy to understand. The ideas are presented in a clear and understandable manner. The abstract contains a significant amount of information and findings. Please note that the abstract writing style in the journal does not contain headings but is written in paragraph format.
3. The introduction directly addresses the topic. There is a clear and concise statement of the problem. There is a clear outline of the main variables of the study. The formulation of the research problem in a scientifically sound manner. However, research problem is well-written, the research gap is not clearly defined. It is imperative for the authors to clarify how their research contributes to the filling of a theoretical, conceptual, and/or methodological gap.
4. I recommend updating References in the introduction section. Out of 27 references; 7 are between 2020 to 2023 only. Older publications may be limited. Particularly those that date back to 2017 and before.
5. Researchers have set specific, clear, and straightforward objectives underlying their research that are formulated in a straightforward manner in order to make the research more meaningful. The application of research in the area of specialization plays an essential role.
6. In the study, a number of relevant studies have been found to have been incorporated in order to provide a comprehensive picture. authors have used and incorporated the findings of the previous studies as well as drawn on them in order to formulate study hypotheses. Again older publications may be limited. Particularly those that date back to 2017 and before.
7. Research methodology takes into account the study's objectives. In accordance with scientific standards and guidelines, systematic methods have been applied.
· However, authors must clarify more about HOW the platform specified the seven hundred eligible participants and HOW they did the sample frame. Sampling techniques need more clarification.
· Furthermore, In Section 3.2 of the Methodology, "Table 2. Constructs and measurement items in the questionnaire", I recommend moving the measurement instrument to the Appendices.
8. The results of the study are presented in an organized and accurate manner. The study's findings are based on sound scientific principles. It was found that the study's results were in agreement with those found in prior research and theories. Although the authors cite recent publications, older publications may be limited. However, I suggest adding a results summary table for all the tested hypotheses. The table headings below can be followed.
Table --. Results summary.
No. |
Hypothesis |
Beta |
t Statistic |
Decision |
9. In the discussion section, please move sub-sections 5.2. Implications and 5.4. Limitations and future work (lines 415-478) to a new section “conclusion section.
10. Authors must add conclusion section reflect clearly the research objectives and is it novel.
· Provide a summary of your findings.
· Synthesize the most important points.
· Highlight the study's key findings.
· Identify the remaining problems and questions.
I. study limitations
II. Provide a direction for the future.
III. Ensure that your final statement is strong and concise.
Throughout the research and at the end, references were used to provide a solid scientific foundation. In the body of the paper, references are limited to those that are cited in the text and in the references.
Additionally, the style of writing engages the reader and helps to create a feeling of creativity in him or her as a result of reading it. It is clear throughout the entire study that the parts are interconnected in a clear and logical manner.
Finally, I would like to suggest the following useful articles to be used to enrich this manuscript (Please note that this is merely a suggestion and is not mandatory to add)
i. "The Use of a Technology Acceptance Model (TAM) to Predict Patients’ Usage of a Personal Health Record System: The Role of Security, Privacy, and Usability." International Journal of Environmental Research and Public Health 20.2 (2023): 1347.
ii. "Exposure Detection Applications Acceptance: The Case of COVID-19." International Journal of Environmental Research and Public Health 19.12 (2022): 7307.
iii. "Risk of fear and anxiety in utilising health app surveillance due to COVID-19: Gender differences analysis." Risks 9.10 (2021): 179.
iv. "Counseling for Health: How Psychological Distance Influences Continuance Intention towards Mobile Medical Consultation." International Journal of Environmental Research and Public Health 20.3 (2023): 1718.
v. "Predictors of patients’ acceptance of video consultation in general practice during the coronavirus disease 2019 pandemic applying the unified theory of acceptance and use of technology model." DIGITAL HEALTH 9 (2023): 20552076221149317.
vi. "Dr. Google: Physicians—The Web—Patients Triangle: Digital Skills and Attitudes towards e-Health Solutions among Physicians in South Eastern Poland—A Cross-Sectional Study in a Pre-COVID-19 Era." International Journal of Environmental Research and Public Health 20.2 (2023): 978.
vii. "Understanding the Antecedents and Effects of mHealth App Use in Pandemics: A Sequential Mixed-Method Investigation." International Journal of Environmental Research and Public Health 20.1 (2023): 834.
I recommend that this paper be accepted with major revision.
Author Response
Response to Reviewer #2
It is possible to consider your paper entitled " Modelling consumer acceptance and usage behaviors of m- health: An integrated model of Self-Determination Theory, Task-Technology Fit and Technology Acceptance Model" for publication only after some revision. I have listed my comments below.
Comment #1
From a scientific perspective, the title is well-formulated.
Response:
Thanks for your recognition. Very much appreciated.
Comment #2
The abstract is well organized and easy to understand. The ideas are presented in a clear and understandable manner. The abstract contains a significant amount of information and findings. Please note that the abstract writing style in the journal does not contain headings but is written in paragraph format.
Response:
The authors appreciate the comment made by the reviewer. We have revised the abstract in paragraph format to meet the requirements of the journal.
Comment #3
The introduction directly addresses the topic. There is a clear and concise statement of the problem. There is a clear outline of the main variables of the study. The formulation of the research problem in a scientifically sound manner. However, research problem is well-written, the research gap is not clearly defined. It is imperative for the authors to clarify how their research contributes to the filling of a theoretical, conceptual, and/or methodological gap.
Response:
The authors appreciate the comment made by the reviewer. We have significantly revised some parts of the introduction section to clearly identify the research gap and to reinforce the rationale of this study (Also see responses to comments 1-3 by reviewer #1). To recap, first, “the troubling problem of non-acceptance or under-use continues to be an essential concern in m-health practice.”(page 2, lines 63-64). Second, as revised in the Introduction part (page 2, lines 62-63), we have provided a brief summary of previous studies on m-health acceptance, as shown in the new Table 1 (Page 2), and identified theoretical research gaps of previous studies. Specifically, we argue that there is a surprising dearth of research on motivational and technical factors that are crucial in understanding consumers’ usage of m-health as follows:
“Although existing studies have examined a set of variables to facilitate m-health acceptance, such as perceptions of m-health (e.g., perceived ease of use, perceived usefulness and perceived privacy risk), social factors (e.g., subjective norms and social influence) and personal factors (e.g., age, gender and experience) (See Table 1 for a summary), there is a surprising dearth of research on motivational and technical factors that are crucial in understanding consumers’ usage of m-health.” (Page 2, lines 70-76)
In addition, the incorporation of SDT and TTF in our research model has been well justified accordingly. In particular, why constructs in SDT and TTF could be effective in developing an understanding of consumers’ usage behaviours in relation to m-health has been well explained. Finally, we identified that “However, few studies have examined consumers’ m-health acceptance from integrated perspectives of technology perceptions, users’ self-motivations, and task-technology fit.” (Page 4, lines 116-117) This leads to our motives to conduct this study. We hope this has appropriately addressed the theoretical and conceptual gaps of existing studies.
As for the methodological gap, we think this is a limitation of the present study, as also presented in a number of previous studies. We agree that this might be a promising future research direction. As we have stated in Section 5.4, we encourage “future studies could consider extending our research in a longitudinal way to examine the changes in user acceptance and usage behaviors of m-health.” (Page 14, lines 512-513)
Comment #4
I recommend updating References in the introduction section. Out of 27 references; 7 are between 2020 to 2023 only. Older publications may be limited. Particularly those that date back to 2017 and before.
Response:
The authors appreciate the comment made by the reviewer. We have updated the references by limiting older ones and including more new references. Now, there are 18 references between 2020 to 2023 in the current reference list.
Comment #5
Researchers have set specific, clear, and straightforward objectives underlying their research that are formulated in a straightforward manner in order to make the research more meaningful. The application of research in the area of specialization plays an essential role.
Response:
Thanks for your recognition. Very much appreciated.
Comment #6
In the study, a number of relevant studies have been found to have been incorporated in order to provide a comprehensive picture. authors have used and incorporated the findings of the previous studies as well as drawn on them in order to formulate study hypotheses. Again older publications may be limited. Particularly those that date back to 2017 and before.
Response:
Thanks for your recognition. Very much appreciated.
Comment #7
Research methodology takes into account the study's objectives. In accordance with scientific standards and guidelines, systematic methods have been applied. However, authors must clarify more about HOW the platform specified the seven hundred eligible participants and HOW they did the sample frame. Sampling techniques need more clarification. Furthermore, In Section 3.2 of the Methodology, "Table 2. Constructs and measurement items in the questionnaire", I recommend moving the measurement instrument to the Appendices.
Response:
The authors appreciate the comment made by the reviewer. As there are a number of m-health platforms in practice, we failed to collect specific data on the platform used by the participants. However, we did collect specific data on participants’ main purpose of using m-health, as shown in Table 2. In addition, we have provided more details on the description of m-health platforms at the beginning of the survey.
“The questionnaire consisted of two sections. The first section gave descriptions of m-health that could facilitate participants’ understanding of the m-health context to be surveyed. A brief description of m-health was provided as follows: “M-health services refer to healthcare services that are delivered through mobile technology (e.g., smartphones, tablets and wearable devices). Representative examples of widely used m-health applications included Chunyu Doctor, Dingdang Medicine Express, and Ping An Good Doctor.” The second section listed items that were used to collect participants’ demographic information and responses to the proposed constructs…”(Page 8, lines 284-291)
In addition, the sampling process are updated and specified.
“Data were collected using Wenjuanxing Survey (www.sojump.com), one of the most popular online survey platform in China, which has been widely used in previous studies. The survey platform owns a sample database of 2.6 million active members with a wide range of demographic characteristics and geographical distributions in China.” (Page 7, lines 262-265)
“The survey platform randomly distributed the questionnaire to 700 eligible participants in their sample database, of which, 623 valid samples were received (response rate: 89%) and used for data analysis.” (Page 7, lines 274-276)
In addition, as suggested by the reviewer, the measurement instrument has been presented in Appendix 1. (Page 18)
Comment #8
The results of the study are presented in an organized and accurate manner. The study's findings are based on sound scientific principles. It was found that the study's results were in agreement with those found in prior research and theories. Although the authors cite recent publications, older publications may be limited. However, I suggest adding a results summary table for all the tested hypotheses. The table headings below can be followed.
Response:
The authors appreciate the comment made by the reviewer. We have updated the references by limiting older references and including more new references. In addition, we have provided a results summary table for all the tested hypotheses (Table 6), which included the tested hypotheses and their statistical results.
Comment #9
9.In the discussion section, please move sub-sections 5.2. Implications and 5.4. Limitations and future work (lines 415-478) to a new section “conclusion section.
Response:
The authors appreciate the comment made by the reviewer. As suggested by the reviewer, we have moved sub-sections 5.2. Implications and 5.4. Limitations and future work to a new section conclusion section (Section 6. Conclusion)
Comment #9
Authors must add conclusion section reflect clearly the research objectives and is it novel.
- Provide a summary of your findings.
- Synthesize the most important points.
- Highlight the study's key findings.
- Identify the remaining problems and questions.
I.study limitations
II.Provide a direction for the future.
III.Ensure that your final statement is strong and concise.
Response:
The authors appreciate the comment made by the reviewer. We have revised the manuscript by reorganizing related sections and including a new conclusions section accordingly. Please kindly refer to Page 12-15 for the conclusions section.
Comment #10
Throughout the research and at the end, references were used to provide a solid scientific foundation. In the body of the paper, references are limited to those that are cited in the text and in the references. Additionally, the style of writing engages the reader and helps to create a feeling of creativity in him or her as a result of reading it. It is clear throughout the entire study that the parts are interconnected in a clear and logical manner.
Response:
Thanks for your recognition. Very much appreciated.
Comment #11
Finally, I would like to suggest the following useful articles to be used to enrich this manuscript (Please note that this is merely a suggestion and is not mandatory to add)
i."The Use of a Technology Acceptance Model (TAM) to Predict Patients’ Usage of a Personal Health Record System: The Role of Security, Privacy, and Usability." International Journal of Environmental Research and Public Health 20.2 (2023): 1347.
ii."Exposure Detection Applications Acceptance: The Case of COVID-19." International Journal of Environmental Research and Public Health 19.12 (2022): 7307.
iii."Risk of fear and anxiety in utilising health app surveillance due to COVID-19: Gender differences analysis." Risks 9.10 (2021): 179.
iv."Counseling for Health: How Psychological Distance Influences Continuance Intention towards Mobile Medical Consultation." International Journal of Environmental Research and Public Health 20.3 (2023): 1718.
v."Predictors of patients’ acceptance of video consultation in general practice during the coronavirus disease 2019 pandemic applying the unified theory of acceptance and use of technology model." DIGITAL HEALTH 9 (2023): 20552076221149317.
vi."Dr. Google: Physicians—The Web—Patients Triangle: Digital Skills and Attitudes towards e-Health Solutions among Physicians in South Eastern Poland—A Cross-Sectional Study in a Pre-COVID-19 Era." International Journal of Environmental Research and Public Health 20.2 (2023): 978.
vii."Understanding the Antecedents and Effects of mHealth App Use in Pandemics: A Sequential Mixed-Method Investigation." International Journal of Environmental Research and Public Health 20.1 (2023): 834.
Response:
We would like to thank the reviewer for providing so many articles for our reference. Some of them are highly related to our study and have been well cited in our study.
Comment #12
I recommend that this paper be accepted with major revision.
Response:
The authors appreciate the comments made by the reviewer. We have tried to address each of the comments and revised the manuscript accordingly, as summarized above, and we trust these have enhanced the quality of the manuscript.

Round 2
Reviewer 1 Report
I would like to thank the author(s) of this study for taking into account all the provided comments when revising the paper. The overall quality of the paper has improved a lot. However, the manuscript will benefit from a quick proofreading session, also it will be helpful to check the quality of provided figures.
Reviewer 2 Report
Modelling consumer acceptance and usage behaviors of m- health: An integrated model of Self-Determination Theory, Task-Technology Fit and Technology Acceptance Model
Dear Authors,
Thank you for giving me a chance to review your manuscript again. It is possible to consider your paper entitled " Modelling consumer acceptance and usage behaviors of m- health: An integrated model of Self-Determination Theory, Task-Technology Fit and Technology Acceptance Model" for publication only after some revision. I have listed my comments below.
· The research problem is well-written, the research gap still is not clearly defined. It is imperative for the authors to clarify how their research contributes to the filling of a theoretical or contextual or methodological gap. Which type of previous gaps this study contributes to cover. Please add it clearly in abstract and in the introduction section in one small paragraph.
· Please add a paragraph at the end of the introduction presenting the sections of the paper and their content for aiding the reader.
· Add a literature review section to provide a wide review on m-health research after introduction section. It is important that this section includes a discussion and explanation of recent and related literature on m-health.
· Moving Table 1. A summary of m-health acceptance studies and their major findings to to this new section “literature review: and before “2. Theoretical background and research hypotheses”. The below listed articles are important. These articles will be useful to enrich this manuscript.
i. "The Use of a Technology Acceptance Model (TAM) to Predict Patients’ Usage of a Personal Health Record System: The Role of Security, Privacy, and Usability." International Journal of Environmental Research and Public Health 20.2 (2023): 1347.
ii. "Exposure Detection Applications Acceptance: The Case of COVID-19." International Journal of Environmental Research and Public Health 19.12 (2022): 7307.
iii. "Risk of fear and anxiety in utilising health app surveillance due to COVID-19: Gender differences analysis." Risks 9.10 (2021): 179.
iv. "Healthcare Workers’ Perspectives of mHealth Adoption Factors in the Developing World: Scoping Review." International Journal of Environmental Research and Public Health 20, no. 2 (2023): 1244.
v. "The impact of using mHealth Apps on improving public health satisfaction during the COVID-19 pandemic: A digital content value chain perspective." In Healthcare, vol. 10, no. 3, p. 479. MDPI, 2022.
vi. "The determinants of user acceptance of mobile medical platforms: An investigation integrating the TPB, TAM, and patient-centered factors." International Journal of Environmental Research and Public Health 19, no. 17 (2022): 10758.
vii. "Factors affecting the perceived usability of the COVID-19 contact-tracing application “Thai chana” during the early COVID-19 omicron period." International Journal of Environmental Research and Public Health 19, no. 7 (2022): 4383.
· Theoretical background and research hypotheses section: Authors must add arguments for every hypothesis. Although previous studies discussed TAM, authors should reinterpret and logically derive hypotheses in accordance with the context of this study (H1, H2 and H3)
· In the data analysis section please clarify which software authors used to analysis data. SEM generally divided into Variance-based SEM / PLS-SEM (e.g. SmartPLS) and Covariance-based SEM (e.g. AMOS which is an extension module from SPSS). If AMOSE, please clarify more about it.
· The authors mentioned in lines 300-301 “The reliability is considered good if Cronbach’s alpha is larger than 0.70; and it is acceptable if Cronbach’s alpha is larger than 0.6.” add references support these values.
· The authors mentioned in lines 301-305 “Convergent validity is considered acceptable, if the composite reliability of the construct is higher than 0.70 and the average variance extracted (AVE) of the construct is larger than 0.50. Discriminant validity is verified if the square root of the AVE for a given construct is larger than its correlations 304 with any other constructs.” add references support these values.
· The authors mentioned in lines 310-313 “A good fit was indicated by the ratio of chi-square to degrees of freedom (χ2 /df < 3), the goodness-of-fit index (GFI ≥ 0.90), the adjusted goodness-of-fit index (AGFI ≥ 0.80), the comparative fit index (CFI ≥ 0.90), the incremental fit index (IFI ≥ 0.90), the Tucker Lewis index (TLI ≥ 0.90), and root mean square error of approximation (RMSEA < 0.08).” add references support these values.
· Figure 2 replace it with the original figure from AMOS including indices.
· Table 6 is not clear. add Standard deviation (STDEV), Beta value, T statistics (|O/STDEV|) or P Value
